# Current Effects of Cyanobacteria Toxin in Water Sources and Containers in the Hartbeespoort Dam Area, South Africa

**DOI:** 10.3390/ijerph16224468

**Published:** 2019-11-13

**Authors:** Matodzi Michael Mokoena, Murembiwa Stanley Mukhola

**Affiliations:** Department of Environmental Health, Tshwane University of Technology, Private Bag X680, Pretoria 0001, South Africa; mukholams@tut.ac.za

**Keywords:** Hartbeespoort Dam, microcystins, water sources, water containers, blooming season, decaying season

## Abstract

The study investigated the effects of cyanobacteria toxins such as microcystins in water sources and water stored in containers during its blooming and decaying seasons. Samples from water sources and containers near the Hartbeespoort Dam in South Africa were analysed using a microcystin ELIZA test kit. Microcystins were present in water sources used by the community, with an average of 4.3 μg/L in communal tap water and 4.8 μg/L in the water stored in tanks. The concentration of microcystins was lower in groundwater in the decaying season (0.38 μg/L) than in the blooming season (1.4 μg/L). Although microcystins were present in the storage containers, the average levels in all water samples were below the acceptable limit of 1 μg/L. The present study confirmed the presence of microcystins in the water storage containers. Therefore, it is suggested that water used for drinking from community water sources should be treated before storage to eliminate microcystins.

## 1. Introduction

Cyanobacteria (blue-green algae) first appeared about 2.5 billion years ago, and their usual habitat is water [1]. These bacteria grow on the surface of water and multiply rapidly to form scum or mats on the surface of the water, which are known as blooms [2]. Cyanobacteria blooming happens in the summer season. This is because of the high temperatures and heavy rainfall, which results in an increase in flooding. During flooding, more nutrients such as phosphorus and nitrogen are washed into the water body. The phosphorus and nitrogen are food for cyanobacteria. 

Approximately 30 species of cyanobacteria have been proven to be associated with toxic water blooms. These bacteria are grouped further according to their biological effects on human systems [3]. However, of greater concern are the three classes of toxins that are produced: Hepatotoxins that attack the liver; neurotoxins that attack the nervous system; and dermatotoxins that cause skin irritation. Oberholster and Ashton [4] reported various toxins and their biological effects on humans; however, the most studied toxins are microcystins, which are known to promote tumours and cancer. The presence of these toxins is assumed to be higher in the summer time or blooming season, however, the toxins in the cells are still active for 21 days after the cell decays. Therefore, the cyanobacteria cell can pose health risks during the dying stage as it releases toxins into the surrounding water. 

Falconer and Humpage [5] reported that microcystins are hepatotoxins that may cause liver damage and promote tumour growth in humans and animals due to the inhibition of protein phosphatase 1. Microcystins are of great concern to human health because they include toxins that are potentially harmful to both humans and animals if consumed [6]. There are more than 80 known microcystin toxins that need to be treated during water purification. 

It has been suggested that most water treatment plants are not capable of treating for cyanotoxins [7,8]. Only the most recently commissioned water treatment plants are reported to be able to treat for microcystins because they use an activated carbon system [9]. The occurrence of toxins in drinking water depends on the level of raw water contamination and the effect of the water treatment used. Many studies conducted so far in South Africa on cyanobacteria have indicated that microcystins are problematic. Most of these studies focused either on the water treatment plant [4,10], wild animals [11] or contamination of different water sources across the country [4]. A study by Fosso-Kankeu et al. [12] assessed the presence of toxins in drinking water at the point of use with the addition of powdered activated carbon as an auxiliary treatment to remove bad taste and odour. Cyanobacteria release toxins due to stress in the blooming season and death in the decaying season. During a cyanobacteria bloom, an excess of dead and decaying cyanobacteria can result in hypoxia or anoxia. When these blooms decay in enclosed coastal environments, they can leach nutrients, organic matter, and water-soluble toxins, which consequently cause localized anoxia, fish kills and mortality in marine organisms [13]. The decay of cyanobacteria also happens when the conditions are unfavorable such as in the winter season due to low temperature, less rainfall and lack of nutrients. The decaying of cyanobacteria has been reported in other countries such Wuxi City where it emits a strong septic odor; this has also been reported in the area around the Hartbeespoort Dam in South Africa [14] This study aimed to assess the effects of cyanobacteria toxins including the concentration levels of microcystin in water containers from different water sources in different seasons.

## 2. Materials and Methods

### 2.1. Study Design

Quantitative methods were used to assess the level of microcystin contamination in drinking water from community water sources and in stored water in containers.

### 2.2. Study Area and Sample Size

The study took place in the communities of Meerhoff, Refentse, Kosmos and Zandfontein, which are situated around the Hartbeespoort Dam in the North West Province of South Africa. Selected study communities use different water types for drinking and other households’ chores. Meerhoff community rely on water from the Rand Water Board and boreholes, while Refentse and Kosmos use tank water supplies sourced from the Schoemansdal water treatment plant and Zandfontein households depend on boreholes and they also use canal water passing by from the dam. The study used a convenience sampling method, which involved the inclusion of households in the study as per the household members’ availability and ability to complete the study. Seventy-six households were systematically selected for the purpose of this study, from different settlements around the Hartbeespoort Dam area. The participants consisted of every member of the selected households. After consent was granted by the household member, water samples were requested from their storage containers and their respective water sources.

### 2.3. Water Sample Collection

All water sources used by the four communities included in the study were sampled in the summer (bloom) and winter (decay) seasons. Two sterile whirl-park sampling bags (500 mL) were used to collect water samples from communal taps (water from the Schoemansdal water treatment plant), groundwater, the Rand Water Board and tankers’ supply. The number of samples from water sources was n = 27 and n = 109 came from water containers; they were collected in both seasons. To stabilize microcystin production and degradation in the samples, two drops of Lugol’s solution were added to each water sample, which was immediately placed in a black plastic bag to prevent exposure to sunlight [15] and stored at 4 °C in a portable ice chest. Upon arriving at the laboratory, 2 mL of each water sample was decanted into an Eppendorf tube and frozen at −80 °C until further analysis of the toxins.

### 2.4. Microcystin Materials

The following materials were used during analysis of microcystins in water samples. Micro-pipettes with disposable plastic tips (20–200 µL), multi-channel pipettes (50–300 µL), deionized water, paper towels, a timer, tape or parafilm, a microtiter plater reader (wavelength 450 nm) and microtiter plate washer. All of the frozen water samples in Eppendorf tubes were thawed and the temperature was allowed to reach ambient before analyses. Analyses for microcystins were performed with the Abraxis Microcystins-ADDA ELISA kit from Toxisolutions in South Africa, following the Abraxis procedure (PN.520011) as detailed below, the test kit come with the standard, control, sample diluent, antibody, enzyme conjugate, substrate and stop solutions, which are ready to use.

### 2.5. Microcystin Analysis

1. Add 50 μL of the standard solutions, control, or samples (Table 1) into the wells of the test strips according to the working scheme given. Analysis in duplicate or triplicate is recommended.

2. Add 50 μL of the antibody solution to the individual wells successively using a multi-channel pipette or a stepping pipette. Cover the wells with parafilm or tape and mix the contents by moving the strip holder in a circular motion on the benchtop for 30 seconds. Be careful not to spill the contents. Incubate the strips for 90 minutes at room temperature.

3. Remove the covering and decant the contents of the wells into a sink. Wash the strips three times using the 1X wash buffer solution. Please use at least a volume of 250 μL of wash buffer for each well and each washing step. Remaining buffer in the wells should be removed by patting the plate dry on a stack of paper towels.

4. Add 100 μL of the enzyme conjugate solution to the individual wells successively using a multichannel pipette or a stepping pipette. Cover the wells with parafilm or tape and mix the contents by moving the strip holder in a circular motion on the benchtop for 30 seconds. Be careful not to spill the contents. Incubate the strips for 30 minutes at room temperature.

5. Remove the covering and decant the contents of the wells into a sink. Wash the strips three times using the 1X wash buffer solution. Please use at least a volume of 250 μL of wash buffer for each well and each washing step. Remaining buffer in the wells should be removed by patting the plate dry on a stack of paper towels.

6. Add 100 μL of substrate (color) solution to the individual wells successively using a multichannel pipette or a stepping pipette. Cover the wells with parafilm or tape and mix the contents by moving the strip holder in a circular motion on the benchtop for 30 seconds. Be careful not to spill the contents. Incubate the strips for 20–30 minutes at room temperature. Protect the strips from sunlight.

7. Add 50 μL of stop solution to the wells in the same sequence as for the substrate (color) solution using a multi-channel pipette or a stepping pipette.

8. Read the absorbance at 450 nm using a microplate ELISA photometer within 15 minutes after the addition of the stopping solution.

### 2.6. Statistical Analysis

Data were captured using Microsoft Excel Office 2010, and statistical analyses were carried out using SPSS V21 (IBM, New York, USA). Different water sources are used in the communities around Hartbeespoort dam and some of these are not resourced from the dam. Therefore, the quality of the water used in terms of microcystins will be different due to the level of exposure to contaminated Hartbeespoort dam water and the type of water treatment process used per source. Thus, the microcystin concentration in the water source (dam, communal tap and tankers’ supply, ground water and Rand Water board) samples were measured. The level of microcystins were also assessed in per blooming and decaying seasons. Similar analyses were done from water samples from storage containers grouped by their respective water sources. The analysis of variance (ANOVA) was used to compare the water qualities among the sources and container samples, where the mean and confidence intervals were used to report the results.

### 2.7. Ethical Consideration

This study did not involve any human or animal tissue, however, ethical consent was required due to the nature of study as it involved permission to collect water samples from families’ stored water containers and family members were interviewed about the source of container samples and to determine if they treated the stored water in their household. The project was submitted to the Tshwane University of Technology (TUT) Ethics Committee and permission was obtained, ref: REC/2012/03/001.

## 3. Results

### 3.1. Microcystins

The primary water sources used in the study area are treated tap and tank water, which are sourced from the Hartbeespoort Dam and treated at the Schoemansdal water treatment plant. The water in the dam is contaminated by cyanobacteria and the treatment plant is not well equipped to treat cyanobacteria toxins. While the groundwater is from the boreholes that people have drilled themselves in their backyard, for the purposes of this study it was assumed that these were contaminated by water from the Hartbeespoort dam. The Rand water supply from the Vaal Dam was the only water that had no link to Hartbeespoort dam water. In South Africa, the guideline for microcystin in drinking water, which were adapted from microcystin LR [16], is less than 1 µg/L. Any microcystins above 1 µg/L are deemed to make the water inacceptable for drinking purposes. Water sample analysis was done as explained in the methodology with control and standards for all analysis of microcystin in water sources and containers.

### 3.2. Microcystins in Water Sources

Figure 1 shows the microcystin contamination of the water sources, grouped by the respective seasons. In most of the dam-water samples, the concentration of microcystin was above the acceptable drinking water guideline (1 µg/L) in both the decaying and blooming seasons, with a mean concentrations of (2.40 μg/L; 95% CI 1.27–2.93) and (1.85 μg/L; 95% CI 0.85–2.06). There were decreases in microcystin concentration from the blooming to the decaying season. The levels of microcystin concentration in the blooming season were (4.33 μg/L; 95% CI 4.19–9.10) and in the decaying season they were (5.00 μg/L; 95% CI 5.18–6.78). The outcomes for the dam-water samples were expected because the dam experiences cyanobacteria blooms as reported by Mokoena et al., 2016 [17]. However, the high number of toxins that was also observed in the decaying season was not expected. This is because increases in cyanobacteria toxins are believed to be higher in the blooming season. The communal tap-water samples were contaminated by microcystins even after treatment by the treatment plant, in both the blooming and decaying seasons (1.75 μg/L; CI 95% 1.27–2.93) and (0.68 μg/L; CI 95% 0.85–2.06, respectively). In the blooming season, the groundwater contained a mean microcystin concentration of 1.42 μg/L at a 95% confidence interval (1.14–2.85 μg/L) while in the decaying season, the groundwater contained a very low concentration of microcystins, with a mean of 0.38 μg/L and a 95% confidence interval of 0.00 μg/L. Since the groundwater was sampled close to the dam, it is possible that the groundwater was contaminated by water from the dam. This contamination may have originated from the surface of the dam.

In Figure 1, the dotted line = acceptable level limit of 1 µg/L [16], Dam Water—water used by the community direct from the dam or from the dam’s canals, Ground Water—groundwater samples, Rand Water Board—water supplied through the tap from the Rand Water treatment plant, Vereeniging, and Tank and Tap water are samples from the Schoemansdal water treatment plant.

The water supply from Rand Water was more affected by microcystins (the mean concentration was 1.80 μg/L) at a 95% confidence interval (2.61–4.43 μg/L) in the blooming season than in the decaying season, as noticed in the tap water sourced from the Hartbeespoort Dam.

### 3.3. Microcystins in Water Containers

The quality of water stored in containers is influenced by the quality of the water sources as well as household hygiene practices. The blooming of cyanobacteria in the surface water of the Hartbeespoort Dam influences the quality of the water stored in containers. Figure 2 shows the results for microcystin found in water containers. Groundwater and Rand Water samples stored in containers show a mean microcystin concentration of less than 1 μg/L in the blooming season (0.001 and 0.21 μg/L) and in the decaying season (0.001 and 0.001 μg/L).

There were some outlier samples (represented as ◦ *) with microcystin concentrations above or below 1 μg/L; these were found mostly in water samples from containers storing water from water tanks, tap water and Rand Water. Outliers were observed more frequently in the decaying season (<0.001 μg/L; CI 95% −0.72 to 0.32; <0.02 μg/L; CI 95% −0.19 to 1.08; and −0.93 μg/L; CI 95% −0.24 to 5.18) than in the blooming season (0.01 μg/L; CI 95% −0.58 to −0.06; 0.39 μg/L; CI 95% −0.49 to 0.57; and −2.20 μg/L; CI 95% −2,40 to −0.62). The tap and tank water were sourced from the same treatment plant. However, the difference was that the tap water was supplied using the water system infrastructure, and the tank water was supplied using trucks, with the water in tanks being stored in a central place in the community. Tanks are filled with water once or twice a week or even once a month. During the period in which the tanks are empty, the sun, dust and other animals such as birds introduce contaminant to the inside of the tanks. There was no reports of the communal tanks being cleaned before being filled with water. The contaminant, and other parameters such as high temperature can catalyse the growth of cyanobacteria cells that were not treated during the water treatment process, as the cyanobacteria is not spread by wind. This increases the microorganisms that re-contaminate the clean water. Groundwater and Rand Water samples were less contaminated in both the blooming season (0.001 μg/L; CI 95% −2.42 to −0.84 and −2.20 μg/L; CI 95% −2.40 to −0.64, respectively) and the decaying season (−1.49 μg/L; CI 95% −1.16 to 1.09 and −0.93 μg/L; CI 95% −0.24 to 5.18, respectively). This suggests that the contamination of water in the containers did not originate from their respective water sources, however, the use of cyanobacteria-contaminated containers causes the contamination of drinking water in the containers.

## 4. Discussion

Cyanobacteria toxins in drinking water pose health risks. Hoffman [10] and Duy et al. [18] have reported that exposure to low concentrations of microcystins for long periods is associated with chronic health problems that take some time to develop. The toxins are released by the cells in both the blooming and decaying season.

Hartbeespoort Dam is one of several dams in South Africa that are highly contaminated by cyanobacteria, which release toxins. The dam is also a major water source in the North West province and is used for irrigation purposes. Cyanobacteria toxins and their respective cells are reduced or removed during water treatment, especially by chlorinated water treatment as it degrades the toxins. Cyanobacteria blooming during the summer can be reduced by managing the flow of the nutrients into the dams that provide food to Cyanobacteria. To reduce the nutrients in South African rivers and dams, especially in Gauteng province, will require control of the waste water flowing into the rivers. As [19] reported, waste stabilization ponds (WSPs) are usually used to provide an effective and low cost means of handling domestic wastewater in smaller towns and communities. The use of WSPs has advantages over conventional waste water treatment plants because they are very simple to design, operate, and maintain, and do not necessarily need skilled manpower. Waste water collection and treatment management are the critical key to prevent black-water in rivers. Furthermore, building wetlands can trap the contents of wastewater. It has also been suggested that hyacinths are able to control the blooming of Cyanobacteria as they use all the nutrients in the water, which will affect the growth of Cyanobacteria [20].

Some residents using tank water reported that the water supply was not constant, which resulted in 2–4 weeks without tank water. The length of time in which water was stored in the containers was not assessed but there is evidence that people normally top up bigger containers without changing the water [21]. In the study area, clean water will be stored and used for drinking and cooking purposes and changed when new water is supplied in the tanks. Because of this problem, residents are often compelled to use other available water sources such as groundwater collected from neighbouring farms, using the same containers that were employed for tank water collection. In addition, residents from an informal settlement indicated that they used the same containers to collect drinking water from the canal as well as water for other household chores such as laundry and bathing. This has an impact on cross-contamination as canal water increases the amount of nutrients in the water container and promotes biofilm growth. Biofilm is a group of microorganisms that grow on the surface of the container, including cyanobacteria.

Contamination of groundwater stored in containers was exacerbated by cross-contamination because the same containers were reported to be used when collecting water for laundry and bathing from other water sources that were contaminated. Most households used light-penetrating, small screw-cap containers. These types of containers have been reported to contribute to cyanobacteria growth and biofilm formation [12]. Even if the concentration of microcystins in the water stored in containers is low, the fact that toxins are produced in the containers poses a serious health risk to the community members using them. The presence of microcystins in drinking water at household level poses a health risk to the immune compromised, the elderly and children less than 5 years old. Moreover, long-term exposure, even to low concentrations of microcystins, can result in chronic health risk for all people. The collection and storage practices regarding water in containers proved to aid the incubation of cyanobacteria cells. Figure 1 shows that despite the water being treated, there were still microcystin concentrations above the acceptable level in both the blooming and decaying seasons. The groundwater, Rand Water and tank water sources demonstrated microcystin concentrations that were lower than the acceptable limit or were undetectable.

## 5. Conclusions

The quality of the water sources did not change much between the seasons in terms of microcystin contamination, which was above the acceptable limit (1 μg/L). There was no significant difference (*p* > 0.05) between the blooming season and the decaying season regarding the microcystin contamination of the water stored in containers. Containers were also shown to be incubators for the production of microcystins since microcystins were found in all the water samples that were stored in containers, regardless of the water quality of the source. Point-of-use treatment practices should be stressed in those areas where water is collected and stored in containers at the household level.

## Figures and Tables

**Figure 1 ijerph-16-04468-f001:**
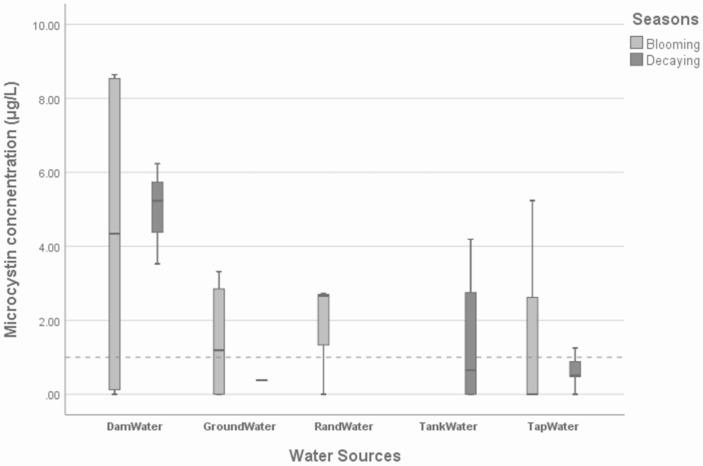
Microcystins in water source samples grouped according to point of collection.

**Figure 2 ijerph-16-04468-f002:**
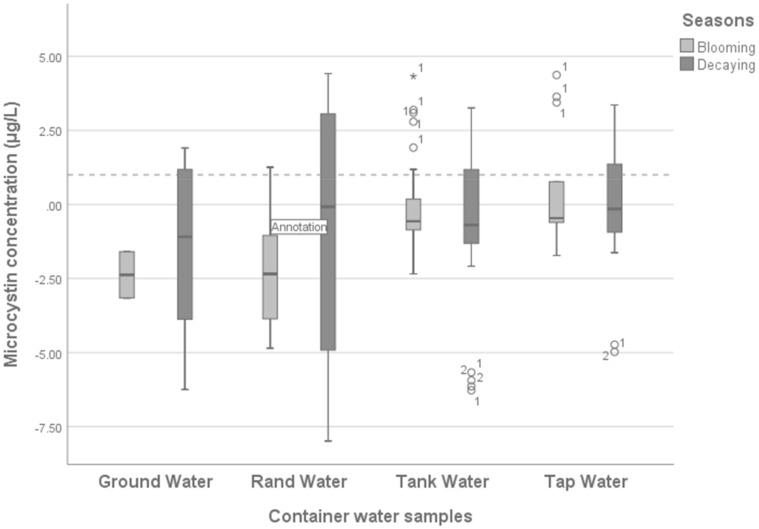
Microcystin concentration in water containers grouped by the sources versus seasons.

**Table 1 ijerph-16-04468-t001:** Standard template used for analyses of microcystin Abraxis plate.

Microcystin Template				
	1	2	3	4
**A**	STD 1 0	STD5 2.0000	SPL2	SPL6
**B**	STD 1 0	STD5 2.0000	SPL2	SPL6
**C**	STD2 0.1500	STD6 5.0000	SPL3	SPL7
**D**	STD2 0.1500	STD6 5.0000	SPL3	SPL7
**E**	STD3 0.4000	CONTROL	SPL4	SPL8
**F**	STD3 0.4000	CONTROL	SPL4	SPL8
**G**	STD4 1.0000	SPL1	SPL5	SPL9
**H**	STD4 1.0000	SPL1	SPL5	SPL9

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
