# Peer review of "Current Effects of Cyanobacteria Toxin in Water Sources and Containers in the Hartbeespoort Dam Area, South Africa"

_ijerph, 2019, doi:10.3390/ijerph16224468_

Round 1
Reviewer 1 Report
The manuscript has investigated the concentrations of cyanobacteria toxin in water sources and water containers in blooming and decaying seasons. It's a meaningful research. They provided basic data for the management and risk assessment of drinking water in local. Some comments are listed below:
The current situation of cyanobacteria bloom in Hartbeespoort Dam should provide in Materials and Methods. It should describe how to sample water sources and water containers. Statistical method in Line 77 should be SPSS. The unit of microcystins concentration should give in the vertical axis in Figure 1 and 2. Solid line of 1 μg/L didn’t show in Figure 1. The labels in the horizontal axis should describe clear. The circle and data in Figure 2 should describe in detail. Figure 1 didn’t support the data in Line 95-Line 97. There are some samples with microcystin concentrations below 0 in containers. Does it mean they are below the detection limit?Author Response
Response to Reviewer 1 Comments
Point 1: The manuscript has investigated the concentrations of cyanobacteria toxin in water sources and water containers in blooming and decaying seasons. It's a meaningful research. They provided basic data for the management and risk assessment of drinking water in local. Some comments are listed below:
The current situation of cyanobacteria bloom in Hartbeespoort Dam should provide in Materials and Methods. It should describe how to sample water sources and water containers.
Response 1: The procedure used to sample the water samples from the water sources and containers where described line 78 - 87
Statistical method in Line 77 should be SPSS.
Response 2: Yes the SPSS was change see 131
The unit of microcystins concentration should give in the vertical axis in Figure 1 and 2. Solid line of 1 μg/L didn’t show in Figure 1.
Response 3: both graphs have been included the unit for microcystin concentration (µg/L) line 174 and 191
The labels in the horizontal axis should describe clear.
Response 4: the labelling were updated
The circle and data in Figure 2 should describe in detail.
Response 5: yes this has been describe as you see paragraph 194 -207
Figure 1 didn’t support the data in Line 95-Line 97. There are some samples with microcystin concentrations below 0 in containers. Does it mean they are below the detection limit 

Response 6: Mostly water samples from the dam were contaminated above acceptable line as seen with dotted line? Yes there were some below 1 ug/L as can be seen in blooming season the lower confident interval was 0,85.
Reviewer 2 Report
This is a very interesting and informative manuscript. It deals with an important topic. It is well organized and well written. However, there is a minor thing that needs to be explained before it can be accepted for publication in the International Journal of Environmental Research and Public Health.
On lines 135 and 136
The reader would like to know, besides contamination from the sources (the dam), is there a possibility of additional contamination from others sources during the handling, storage and the use of water. Therefore, it will be more useful, if the authors explain more the situation of the tanks i.e. are they elevated tanks or are they built on the ground, are they covered or not, how the withdrawal of water from tanks takes place, do people use water from a side tap or are they filling their containers with water from an open in the tank, are these tanks managed by the authorities or committees from the community which might reduce further contamination or not?
Author Response
On lines 135 and 136
The reader would like to know, besides contamination from the sources (the dam), is there a possibility of additional contamination from others sources during the handling, storage and the use of water. Therefore, it will be more useful, if the authors explain more the situation of the tanks i.e. are they elevated tanks or are they built on the ground, are they covered or not, how the withdrawal of water from tanks takes place, do people use water from a side tap or are they filling their containers with water from an open in the tank, are these tanks managed by the authorities or committees from the community which might reduce further contamination or not?
Response 1: Paragraphs 202 – 207 and 234 – 242 addresses why is the different of the water quality from Tap to Tank water while water were sourced from the same treatment plant
Reviewer 3 Report
Review Comments
Manuscript ID: ijerph - 628181
This manuscript presents some informative findings on a study of microcystins in raw and stored water samples. The study design is lacking details and adequate planning with regard to sample collection and analysis. Field blank samples should be included in such study to account for important water quality parameters, storage, and handling variables in the collected water samples. As discussed by the authors, the findings are important in terms of public health significance and water storage and handling on consumers' end. It is suggested that the current study be revised and evaluated to better plan the sampling, analysis, and reporting protocols for the study. This will help with improving the study design and provide for more informative and conclusive presentation of findings. Therefore, I recommend holding off on publication of the current manuscript, pending additional data gathering and major revisions to the current manuscript.
Below are my specific comments and revisions for the current manuscript:
Line 27-28: Revise statement. It does not read clearly.
Line 54-56: Provide more details on past studies about duration of blooming and decaying seasons for cyanobacteria and their toxins release.
Line 62- 63: Specify how the 76 household samples were distributed among the targeted commnities.
Lines 65-66: Specify what "both seasons" were. Explain what "Rand Water" is.
Line 74: Explain in more details. What did the standard and control solutions consist of? Was there replication of samples?
Line 77: Clarification is needed. What water samples were compared to each other? Stored water compared with source waters? What statistical methods were used?
Line 82-83: What data was collected from interviews with household members? How were collected interview data were used in this study?
Line 88 and Figure 1: Source water descriptions are unclear. Is the water taken from Hartbeesport the water source for all samples, except groundwater, shown on Figure 1? How are the samplng locations related to each other? Where did the groundwater sample come from in relation to the other sampling locations. These could be illustrated on a map of sampling locations and/or a summary table.
Figure 1: The caption should indicate what data the box plots present. Also, it is not clear how many samples were collected at each location. Is N = 33 total number of samples from all locations? The solid line referred to as the limit of 1 micrgram per liter appears to be missing on the figure.
Figure 2: The caption for this figure is unclear. The figure shows three box plots for groundwater samples. It is not clear if these are from same groundwater source, but from different storage containers? "Blooming" concentration data box plot is not shown for one of the two groundwater samples. It is also not clear what the numbers by outliers circles on box plots for tank and tap waters represent; and what the asterisks on tank water box plot represents.
Line 119: This section should also present data on control samples. Were the control blank samples included in the study for comparisons regarding stored water quality and microcystin growth?
Line 161: Literature source should be cited in support of the statement on the suggested use of Hyacinths to aid in uptake of nutrients.
Line 163: Indicate what the typical length of water storage was in the studied households.
Line 169 - 186: No chlorine residual data were discussed or reported in the study. If available, water quality parameters, such as free and combined residual chlorine, pH, temperature, and nutrient contents should be discussed. Otherwise, these should be discussed in the context of study limitations.
Author Response
Below are my specific comments and revisions for the current manuscript:
Line 27-28: Revise statement. It does not read clearly.
Response1: The statement was revised 26 - 29
Line 54-56: Provide more details on past studies about duration of blooming and decaying seasons for cyanobacteria and their toxins release.
Response 2: Detail of decaying cyanobacteria has provided, line 54 - 60
Line 62- 63: Specify how the 76 household samples were distributed among the targeted commnities.
Response 3: See Line 68 - 79
Lines 65-66: Specify what "both seasons" were. Explain what "Rand Water" is.
Response 4: Specification and explaination are done in line 81 - 89
Line 74: Explain in more details. What did the standard and control solutions consist of? Was there replication of samples?
Response 5: Table 1 shows the standard and control for the ELIZA test kit in which control is deionised water which assist the research to determine whether worked sterile. All samples as in table 1 are double.
Line 77: Clarification is needed. What water samples were compared to each other? Stored water compared with source waters? What statistical methods were used?
Response 6: Line 132 - 142
Line 82-83: What data was collected from interviews with household members? How were collected interview data were used in this study?
Response 7: line 144 - 149
Line 88 and Figure 1: Source water descriptions are unclear. Is the water taken from Hartbeesport the water source for all samples, except groundwater, shown on Figure 1? How are the samplng locations related to each other? Where did the groundwater sample come from in relation to the other sampling locations. These could be illustrated on a map of sampling locations and/or a summary table.
Response 8: Please see line 152 - 162
Figure 1: The caption should indicate what data the box plots present. Also, it is not clear how many samples were collected at each location. Is N = 33 total number of samples from all locations? The solid line referred to as the limit of 1 micrgram per liter appears to be missing on the figure.
Response 9: Reference line in included see figure 1
Figure 2: The caption for this figure is unclear. The figure shows three box plots for groundwater samples. It is not clear if these are from same groundwater source, but from different storage containers? "Blooming" concentration data box plot is not shown for one of the two groundwater samples. It is also not clear what the numbers by outliers circles on box plots for tank and tap waters represent; and what the asterisks on tank water box plot represents.
Response 10: Figure corrected, three box plot were due to different in labelling the sources name. Line 205 clearly state what are those outlier
Line 119: This section should also present data on control samples. Were the control blank samples included in the study for comparisons regarding stored water quality and microcystin growth?
Response 11: As indicated above that all the analysis have standard and control Table 1 the researcher see no reason of including standard and control ready. However, if need be data is available in can included and figure will then presented with those values. As researcher focused on the samples.
Line 161: Literature source should be cited in support of the statement on the suggested use of Hyacinths to aid in uptake of nutrients.
Response 12: Yes see line 241
Line 163: Indicate what the typical length of water storage was in the studied households.
Response 13: See line 242 - 252
Line 169 - 186: No chlorine residual data were discussed or reported in the study. If available, water quality parameters, such as free and combined residual chlorine, pH, temperature, and nutrient contents should be discussed. Otherwise, these should be discussed in the context of study limitations.
Response 14: Data for the point-of-use water treatment using chlorine is with the research therefore for the purpose of this paper Chlorine and pH were removed
Round 2
Reviewer 3 Report
The revised manuscript is much improved. The authors have addressed my earlier review comments.